# The Intersection of Sexual Orientation, Substance Use, and Mental Health: Findings from Hints 5

**DOI:** 10.3390/healthcare12202083

**Published:** 2024-10-18

**Authors:** Saredo M. Bouraleh, Bishwajit Ghose

**Affiliations:** 1School of Epidemiology and Public Health, Faculty of Medicine, University of Ottawa, Ottawa, ON K1A 0A1, Canada; sbour133@uottawa.ca; 2Interdisciplinary School of Health Sciences, University of Ottawa, Ottawa, ON K1A 0A1, Canada

**Keywords:** e-cigarette use, tobacco use, LGBTQ health, mental health, health behaviour

## Abstract

**Objectives:** In this study, we aimed to investigate (1) the association of tobacco and e-cigarette use with sexual orientation (LGBTQ and heterosexual individuals) and (2) the difference in the association of tobacco and e-cigarette use with self-reported depression by sexual orientation. Methods: The data for this study were obtained from the Health Information National Trends Survey (HINTS 5, Cycle 4). Sample participants included 3583 adults (93.87% heterosexuals). We used multinomial regression to measure the relative risk ratios (RRRs) of being a former and current user versus never a user of tobacco and e-cigarettes and binomial regression to measure the odds ratios of depression between the LGBTQ and heterosexuals. Results: Current smoking prevalence is higher among LGBTQ participants (17.3%) compared to heterosexuals (11.0%). The disparity is even greater for e-cigarette use, with 7.3% of LGBTQ participants being current users versus 2.8% of heterosexuals and 24.5% of LGBTQ participants being former users compared to 9.3% of heterosexuals. Compared to heterosexuals, the relative risk ratio of being a current tobacco user among the LGBTQ participants was about 1.75 times higher [RRR = 1.75, 95%CI = 1.16, 2.64], and that of e-cigarette use was about 2.8 times higher [RRR = 2.81, 95%CI = 1.57, 5.05]. Among current e-cigarette users, heterosexual participants had 1.9 percentage points [risk difference = 1.94, 95%CI = 1.20, 3.13] higher probability of depression, whereas among the LGBTQ participants, the risk was about 3.7 times higher [OR = 3.67, 95%CI = 1.06, 12.74]. Conclusions: Our findings conclude that the LGBTQ are more likely to use tobacco and e-cigarettes compared to heterosexuals and that the risk of depression from e-cigarette smoking is more pronounced among the LGBTQ participants.

## 1. Introduction

In recent years, the consumption of e-cigarette products has increased significantly in the United States, accompanied by a rising burden of mental health issues, such as depression and anxiety [1,2,3]. A national study in 2020 revealed that individuals who use e-cigarettes are more likely to report poor self-perceived physical and mental health as fair or poor compared to non-users [4]. However, LGBTQ individuals are more likely to engage in unhealthy lifestyle behaviours, such as tobacco smoking, than non-LGBTQ individuals. Previous studies have shown that LGBTQ individuals are also more likely to use e-cigarettes and other vaping-related products, particularly those experiencing symptoms of depression [5]. Studies have revealed that a higher percentage of LGBTQ individuals use e-cigarettes compared to heterosexual individuals, with 36.5% being former users and 22.3% being current users [6]. This is partly due to the social pressure and discrimination that LGBTQ individuals often face, leading to reliance on substances as an escape from mental distress [7]. The impacts of various psychosocial stressors, such as stigma and discrimination, reduce the likelihood of individuals seeking timely medical care for their health issues [8,9]. This, in turn, can trigger reliance on self-medication strategies, including the use of tobacco and other psychedelic drugs, to cope with the stress [10,11]. Substance abuse is often used as a coping mechanism by individuals dealing with chronic physical and mental health issues, particularly those who lack social support and face discrimination [12,13,14]. A 2016 study highlighted that LGBTQ individuals facing discrimination are more likely (57.4%) to use substances and develop a substance use disorder [15].

The overall prevalence of smoking among U.S. adults has decreased from 42% in 1965 to 14% in 2019, which is attributed to the lower use of tobacco products, mainly in the higher socioeconomic strata [16]. However, tobacco companies have gradually adapted to this trend by targeting lower-income and other vulnerable communities through innovative marketing strategies, including price reduction, through which sexual minorities have been disproportionately impacted [16]. Furthermore, there is evidence of strategies by tobacco companies to infiltrate the LGBTQ community under the guise of philanthropy (such as sponsoring pride marches and street fairs) [6] that can greatly compromise the effectiveness of tobacco control programs. LGBTQ individuals are also more likely to be exposed to and interact with tobacco-related messages on social media, which could potentially lead to higher tobacco use [17,18].

Tobacco consumption is particularly concerning among the LGBTQ community due to its link with poor mental health outcomes, to which this population is especially vulnerable. A literature review in the Journal of Interdisciplinary Health Sciences highlighted the unique challenges faced by sexual minorities, which increase their vulnerability to substance abuse, leading to poor mental health [19]. Higher consumption of tobacco and smokeless tobacco products such as e-cigarettes are important determinants of psychiatric comorbidities such as stress, anxiety, depression, and suicidal behaviour. Studies have shown that both tobacco [20] and e-cigarette [21] use are associated with an increased risk of suicidal behaviours. Furthermore, studies indicate that e-cigarette use can exacerbate symptoms of depression, particularly in vulnerable populations [2,22,23]. Moreover, recent research has identified a significant association between e-cigarette use and depressive symptoms among LGBTQ adults [5]. From this perspective, tobacco use among sexual minorities is a serious public mental health issue, especially for the LGBTQ community, and more so, as they also face higher levels of health disparities in accessing care and being able to afford the services.

Studies emphasize the increased vulnerability of LGBTQ individuals to tobacco-related health issues, including those linked to e-cigarette use [24,25]. Targeted interventions are necessary to address these inequalities. Despite concerns, there is a dearth of evidence regarding the difference in the risk of depression resulting from smoking and e-cigarette use between sexual minorities and heterosexuals. This study aims to contribute to the existing literature by examining the relationship between tobacco and e-cigarette use (including ever, former, and current use) among LGBTQ individuals and heterosexuals. Additionally, it aims to determine whether the risk of experiencing depression symptoms as a result of tobacco and e-cigarette use differs on the basis of sexual identity.

## 2. Methods

### 2.1. Data Source

The data for this study were obtained from the Health Information National Trends Survey (HINTS) 5, Cycle 4. HINTS is a cross-sectional survey designed to collect data on the American public’s knowledge, attitudes, and behaviours regarding cancer-related information. The survey employs a complex sampling design to ensure national representativeness, including a stratified random sampling of households with an oversampling of minority populations. The data collection methods involve self-administered questionnaires, both in paper form and online, to reach a broad demographic. HINTS 5, Cycle 4 was conducted from February 2020 to June 2020, and it included variables on sexual orientation, substance use, and mental health, among other health-related behaviours and perceptions.

### 2.2. Variable Description

The variables included in this analysis were as follows: Sexual Orientation: Heterosexual and Others (including gay, lesbian, bisexual, and other non-heterosexual orientations); Age Group: 18–34, 35–39, 40–44, and 45+; Race: Non-Hispanic White, Non-Hispanic Black or African American, Hispanic, Non-Hispanic Asian, and Non-Hispanic Other; Marital Status: Married, Divorced, and Single; Education: Less than High School, High School Graduate, Some College, and College Graduate or More; Income Sufficiency: Living comfortably, Getting by, Finding it difficult, and Finding it very difficult; Area: Metropolitan, Micropolitan, Small town, and Rural; Smoking Status: Current smoker and Former smoker; E-cigarette Use Status: Current user and Former user. For additional details on demographic variables and their relationship with sexual orientation, see the Appendix A.

### 2.3. Statistical Analysis

At first, descriptive statistics were carried out to summarize the percentage of current and former tobacco and e-cigarette use across different demographic groups. Chi-square tests were used to assess the statistical significance of these bivariate associations. Next, we estimated the multivariate association between sexual orientation and the use of tobacco and e-cigarettes. Given the multinomial nature of the outcome variable, we used multinomial logistic regression methods for this analysis while controlling for the sociodemographic factors. Specifically, we calculated the relative risk ratios (RRRs) of being a current or former smoker (or e-cigarette user) versus never having smoked (or used e-cigarettes). This step was carried out separately for heterosexual and LGBTQ participants as well. Additionally, we performed binary logistic regression analysis to assess the relationship between smoking and e-cigarette use status and the risk of depression. Separate analyses were conducted for the full sample, heterosexual participants, and LGBTQ participants to explore potential differences while adjusting for the sociodemographic factors. All statistical analyses were performed using Stata 16 (Stata Corp, College Station, Texas). Statistical significance was set at *p* < 0.05 for all analyses.

## 3. Results

### 3.1. Descriptive Analysis

The descriptive analysis shows notable differences in smoking and e-cigarette use across various demographic groups (Table 1). A total of 11.4% of the participants reported smoking currently, while 25.0% reported smoking formerly. Regarding e-cigarettes, 3.1% of the participants reported smoking currently, while 10.2% reported smoking formerly. Chi-square tests revealed significant differences in smoking and e-cigarette use across sexual orientation, age group, race, marital status, education level, and income sufficiency, as indicated by the *p*-values in Table 1.

Among heterosexuals, 11.0% are current smokers, and 25.3% are former smokers (Figure 1). In contrast, the percentage of current smokers is higher among LGBTQ participants at 17.3%. For e-cigarette use, the difference is even more pronounced: 2.8% of heterosexuals are current users compared to 7.3% among LGBTQ participants, and 9.3% of heterosexuals are former users compared to 24.5% among LGBTQ participants.

### 3.2. Association between Sexual Orientation and Use of Tobacco and E-Cigarette

As shown in Table 2, multinomial logistic regression analysis indicates that LGBTQ participants had a higher relative risk ratio of being both current [RRR = 2.81, 95%CI = 1.57, 5.05] and former [RRR = 2.82, 95%CI = 1.95, 4.08] e-cigarette users. Regarding tobacco use, binary logistic regression analysis revealed that the odds ratios were higher than current use only (OR = 1.75, 95%CI = 1.16, 2.64).

### 3.3. Association between Depression and Use of Tobacco and E-Cigarette Stratified by Sexual Orientation

Table 3 shows the odds ratios, based on logistic regression models, of depression by smoking status among heterosexual and LGBTQ participants. It is evident that the odds ratios of depression were noticeably higher among LGBTQ participants (OR = 3.67, 95%CI = 1.06, 12.74) compared to the heterosexual participants (OR = 1.94, 95%CI = 1.20, 3.13). Similarly, for former users of e-cigarette use, the odds ratios of depression were noticeably higher for LGBTQ participants (OR = 1.87, 95%CI = 1.39, 2.50) compared to the heterosexual participants (OR = 3.49, 95%CI = 1.52, 8.00).

## 4. Discussion

The findings indicate significant differences in both tobacco smoking and e-cigarette use between heterosexual and LGBTQ participants. Specifically, LGBTQ participants had a noticeably higher percentage of smoking (17.3%) compared to heterosexuals (11.0%). Similarly, the use of e-cigarettes is more pronounced among LGBTQ individuals, with 7.3% currently using e-cigarettes compared to 2.8% among heterosexual individuals. The higher rates of tobacco and e-cigarette use in the LGBTQ community may be reflective of broader social and psychological dynamics. LGBTQ individuals often face additional societal pressures and stressors, such as discrimination, identity struggles, and social exclusion. These factors can lead to increased stress levels, which might result in greater reliance on smoking and e-cigarette use as coping mechanisms. However, it is important to consider other individual factors that may also contribute to these behaviours. The existing scientific literature supports these findings. For example, minority stress theory suggests that the chronic stress experienced by minority groups, including LGBTQ individuals, contributes to adverse health behaviours such as smoking. Research shows that LGBTQ individuals experience higher levels of minority stress due to stigma and discrimination, leading to increased smoking rates as a form of stress relief. The pronounced disparity in e-cigarette use is particularly notable. LGBTQ individuals are significantly more likely to be current e-cigarette users (7.3%) than heterosexuals (2.8%). One possible explanation is that e-cigarettes might be perceived as a more acceptable or accessible alternative within the LGBTQ community. Studies indicate that the marketing strategies of e-cigarette companies often target LGBTQ individuals, portraying e-cigarettes as a trendy and safer alternative to traditional smoking. This targeted marketing, combined with the community’s unique stressors, may contribute to higher e-cigarette use rates. Economic factors, such as the affordability of e-cigarettes, could also play a role in their increased prevalence among LGBTQ individuals [26]. The higher percentage of former e-cigarette users among LGBTQ individuals (24.5%) compared to heterosexuals (9.3%) could indicate that while LGBTQ individuals are more inclined to try e-cigarettes, they may also decide to quit using them after some time. This pattern might suggest that although e-cigarettes are initially attractive, their long-term use is not sustained, possibly due to emerging health concerns, the realization of their addictive nature, economic factors, or other contextual influences.

This research specifically focuses on differences in usage patterns between heterosexual and LGBTQ participants, highlighting the intersectionality of sexual orientation with other significant sociodemographic factors such as age, race, marital status, education, and income sufficiency. This study found significant differences in smoking behaviours and associated risks among LGBTQ participants, particularly in the context of mental health outcomes, such as depression. The analysis reveals notable differences in smoking and e-cigarette use based on sexual orientation. Compared with their heterosexual counterparts, LGBTQ individuals presented higher prevalence rates for both current smoking and e-cigarette use. Specifically, LGBTQ participants had a 1.75 times greater risk of being current smokers and a 2.8 times greater risk of being current and former e-cigarette users than heterosexual participants. These findings align with previous research indicating higher rates of tobacco and e-cigarette use among sexual minority populations than heterosexual populations [27,28]. Addressing high levels of tobacco and e-cigarette use and adverse health outcomes among LGBTQ individuals requires the development of diverse and unique tobacco control strategies.

The age-related analysis indicates that younger adults (18–34 years) are more likely to engage in e-cigarette and tobacco use compared to middle-aged adults (35–44), irrespective of sexual orientation. This trend is consistent with studies suggesting that younger age groups are more prone to adopting new tobacco products, including e-cigarettes [29]. However, older adults (45+ years) show the highest percentage of current e-cigarette and tobacco use. Additionally, the data show that older adults are more likely to be former tobacco and e-cigarette users, which may reflect successful cessation efforts among this demographic [30]. Racial and ethnic differences in smoking behaviours were also evident. Non-Hispanic White participants were more likely to be current or former smokers compared to other racial groups, which aligns with the existing literature on racial disparities in tobacco and e-cigarette use [31,32]. Moreover, non-Hispanic Black or African American and non-Hispanic Asian participants had lower rates of current e-cigarette use compared to non-Hispanic White participants. Non-Hispanic White individuals may be more inclined to use e-cigarettes as a method to quit smoking tobacco compared to racialized individuals, as suggested by a recent study [33]. Socioeconomic status (SES) also plays a significant role in the mental health outcomes of e-cigarette and tobacco users. Education level was inversely related to smoking prevalence, with higher educational attainment, college graduate or more, associated with higher rates of former tobacco and e-cigarette use. This trend underscores the protective effect of education on health behaviours, as individuals with higher education levels are more likely to access health information and resources that facilitate smoking cessation [34]. Income sufficiency emerged as a critical determinant of smoking status as individuals reporting higher income sufficiency, living comfortably, and having significantly lower rates of current smoking compared to those with lower income sufficiency, such as getting by, finding it difficult, and finding it very difficult. This association highlights the role of socioeconomic factors in tobacco and e-cigarette use, where financial stress may exacerbate smoking behaviours as a coping mechanism, which can further deteriorate their mental health [35]. The data indicate that individuals with lower SES, characterized by lower income and education levels, are more likely to use e-cigarettes, leading to an increased probability of suffering from depression [36]. Moreover, these individuals may have less access to mental health resources, exacerbating the impact of depression. This study revealed no statistically significant associations between tobacco or e-cigarette use behaviours and area of residence, suggesting that region may have a minor effect on tobacco and e-cigarette use among adults.

One of the most striking findings of this analysis is the elevated risk of depression among e-cigarette users. Both current and former e-cigarette users exhibit significantly greater risks of depression than non-users do. Compared with never users, current e-cigarette users had approximately double the risk of depression, and former users presented a similar increased risk. These findings are consistent with previous studies that have reported an association between e-cigarette use and adverse mental health outcomes [37,38,39]. The mechanisms underlying this association could be multifaceted, including the psychoactive effects of nicotine, which can exacerbate symptoms of depression and anxiety [40]. Additionally, the association between e-cigarette use and mental health issues has been documented in several studies, suggesting that individuals with mental health disorders are more likely to use e-cigarettes as a form of self-medication [41]. The analysis revealed that the risk of depression associated with e-cigarette use is not uniform across all demographic groups. LGBTQ participants who are current e-cigarette users have an even higher risk of depression compared to their heterosexual counterparts. Specifically, among current e-cigarette users, heterosexual participants have a 1.9 times increased risk of depression, whereas LGBTQ participants have a 3.7 times increased risk. This pronounced disparity can be explained by the fact that sexual minority individuals face additional stressors that compound the mental health impacts of e-cigarette use, such as discrimination, social stigma, and internalized homophobia [42,43,44]. Economic difficulties, limited access to mental health care, and other social factors may also contribute to this elevated risk and merit further investigation. Additionally, e-cigarettes may be used as a temporary coping mechanism during stressful periods or significant life events. As these stressors are resolved, some individuals might reduce or discontinue e-cigarette use. Conversely, persistent use can lead to dependence, further complicating mental health and potentially exacerbating depression. This pattern suggests that the relationship between e-cigarette use and depression is complex and influenced by both immediate and prolonged factors.

These findings have important implications for public health interventions and policies. The greater prevalence of tobacco and e-cigarette use among LGBTQ individuals requires specific prevention and cessation programs that consider the unique stressors and social determinants affecting this population. However, this study has several limitations that should be considered alongside its contributions to the literature. First, the use of cross-sectional data prevents causal inferences from being made between the outcome and independent variables. The establishment of temporal relationships demands the need for longitudinal studies. Second, the data are self-reported and therefore are subject to bias that occurs from recall or social desirability, thus potentially causing underreporting or overreporting of smoking behaviours and mental health issues. Categorizing sexual orientation as either heterosexual or other may oversimplify and ineffectively represent the degree of diversity that exists in the LGBTQ community. In addition, residual confounding by unmeasured variables—for example, exposure to discrimination or social support—might influence the observed associations. Finally, the survey did not focus specifically on information about the type and frequency of e-cigarette use. Such details might provide a better understanding of the association between e-cigarette use and mental health. These limitations notwithstanding, the results highlight the need for more targeted public health interventions within the context of the higher prevalence of smoking and mental health comorbidities among LGBTQ individuals.

## 5. Conclusions

The relationships between sexual orientation, sociodemographic characteristics, mental health, and tobacco and e-cigarette use behaviours are very complex. Our results highlight pronounced disparities in substance use and mental health outcomes between LGBTQ individuals and heterosexual adults, with LGBTQ individuals exhibiting higher levels of tobacco and e-cigarette use. The odds of being a current tobacco user are significantly greater among LGBTQ individuals, and e-cigarette use is also much more common in this group. Furthermore, the likelihood of experiencing depression in conjunction with e-cigarette use is much higher for LGBTQ individuals. This underscores the need for public health interventions and support services to address the unique challenges of LGBTQ communities. Policymakers and healthcare providers should prioritize reducing substance use and improving mental health among LGBTQ populations. Safe and inclusive environments would help reduce disparities in health outcomes among LGBTQ communities and increase their overall well-being. Further research is necessary to understand the mechanisms perpetuating these disparities and to develop effective intervention strategies. Additionally, future studies should employ longitudinal designs to better grasp causal relationships and guide the development of interventions to reduce disparities between tobacco users and non-users.

## Figures and Tables

**Figure 1 healthcare-12-02083-f001:**
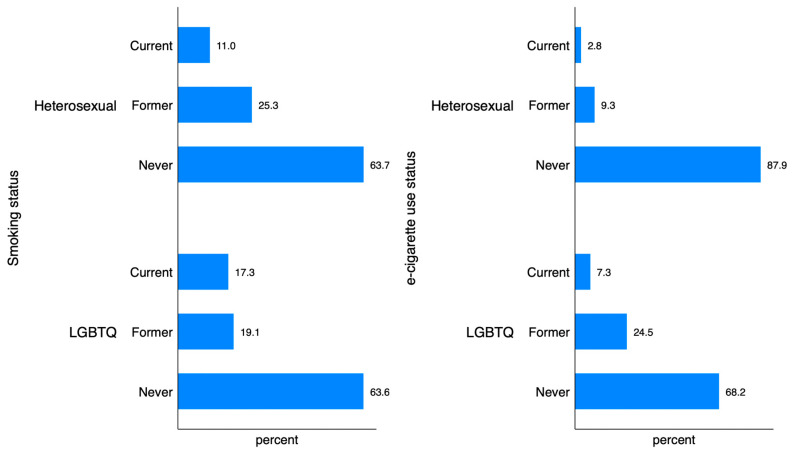
Smoking and e-cigarette use patterns among heterosexual and LGBTQ participants.

**Table 1 healthcare-12-02083-t001:** Descriptive analysis—% of tobacco and e-cigarette use (n = 3583).

	Smoking Status			E-Cigarette Use Status		
	Current (11.4%)	Former (25.0%)	*p*-Value	Current (3.1%)	Former (10.2%)	*p*-Value
**Sexual orientation**						
Heterosexual	90.7 (87.8; 93.5)	95.3 (93.9; 96.7)		85.7 (79.2; 92.2)	85.3 (81.7; 88.9)	
LGBTQ	9.3 (6.5; 12.2)	4.7 (3.3; 6.1)	0.01	14.3 (7.8; 20.8)	14.7 (11.1; 18.3)	0.00
**Age group**						
18–34	11.2 (8.1; 14.3)	6.4 (4.8; 8.0)		30.9 (22.3; 39.5)	31.7 (26.9; 36.5)	
35–39	6.0 (3.7; 8.3)	3.8 (2.5; 5.0)		10.0 (4.4; 15.6)	9.7 (6.7; 12.8)	
40–44	8.7 (6.0; 11.5)	4.6 (3.2; 5.9)		11.8 (5.8; 17.9)	9.7 (6.7; 12.8)	
45+	74.1 (69.8; 78.4)	85.3 (82.9; 87.6)	0.00	47.3 (37.9; 56.6)	48.9 (43.7; 54.1)	0.00
**Race**						
Non-Hispanic White	63.1 (58.3; 67.9)	74.3 (71.3; 77.2)		71.6 (63.1; 80.0)	67.5 (62.6; 72.4)	
Non-Hispanic Black or African American	15.1 (11.5; 18.6)	9.3 (7.4; 11.3)		2.8 (−0.3; 5.8)	10.2 (7.0; 13.3)	
Hispanic	14.3 (10.8; 17.8)	11.5 (9.3; 13.7)		16.5 (9.5; 23.5)	12.7 (9.2; 16.2)	
Non-Hispanic Asian	3.6 (1.8; 5.5)	1.9 (1.0; 2.8)		2.8 (−0.3; 5.8)	3.1 (1.3; 4.9)	
Non-Hispanic Other	3.9 (2.0; 5.8)	3.0 (1.8; 4.1)	0.00	6.4 (1.8; 11.0)	6.5 (3.9; 9.1)	0.00
**Marital status**						
Married	44.4 (39.6; 49.3)	54.6 (51.3; 57.9)		47.3 (38.1; 56.6)	46.8 (41.7; 52.0)	
Divorced	37.0 (32.3; 41.7)	34.5 (31.4; 37.6)		26.8 (18.6; 35.0)	27.7 (23.1; 32.3)	
Single	18.6 (14.8; 22.4)	10.9 (8.9; 12.9)	0.00	25.9 (17.8; 34.0)	25.5 (21.0; 29.9)	0.00
**Education**						
Less than High School	11.6 (8.5; 14.7)	6.1 (4.5; 7.6)		9.8 (4.3; 15.3)	3.3 (1.5; 5.1)	
High School Graduate	24.9 (20.7; 29.1)	19.7 (17.1; 22.3)		17.0 (10.0; 23.9)	18.6 (14.6; 22.6)	
Some College	39.7 (34.9; 44.4)	34.2 (31.1; 37.3)		37.5 (28.5; 46.5)	35.0 (30.1; 39.9)	
College Graduate or More	23.9 (19.7; 28.0)	40.0 (36.8; 43.2)	0.00	35.7 (26.8; 44.6)	43.2 (38.1; 48.2)	0.00
**Income sufficiency**						
Living comfortably	21.6 (17.6; 25.7)	41.0 (37.8; 44.3)		35.7 (26.8; 44.6)	28.5 (23.9; 33.2)	
Getting by	39.7 (34.9; 44.5)	39.3 (36.1; 42.6)		36.6 (27.7; 45.5)	40.7 (35.7; 45.8)	
Finding it difficult	24.6 (20.4; 28.9)	13.7 (11.4; 16.0)		17.9 (10.8; 25.0)	19.4 (15.3; 23.5)	
Finding it very difficult	14.1 (10.7; 17.5)	5.9 (4.4; 7.5)	0.00	9.8 (4.3; 15.3)	11.4 (8.1; 14.6)	0.00
**Area**						
Metropolitan	84.8 (81.3; 88.3)	87.5 (85.3; 89.7)		87.5 (81.4; 93.6)	88.8 (85.6; 92.1)	
Micropolitan	8.8 (6.1; 11.6)	7.6 (5.9; 9.3)		9.8 (4.3; 15.3)	7.4 (4.7; 10.0)	
Small town	4.2 (2.2; 6.1)	2.7 (1.6; 3.7)		2.7 (−0.3; 5.7)	1.9 (0.5; 3.3)	
Rural	2.2 (0.8; 3.6)	2.2 (1.3; 3.2)	0.29	0.0 (0.0; 0.0)	1.9 (0.5; 3.3)	0.44

**Table 2 healthcare-12-02083-t002:** Relative risk ratios of using tobacco and e-cigarette use status between heterosexual and LGBTQ participants.

	Tobacco Use Status	E-Cigarette Use Status
*Current user vs. never*		
Sexual orientation (reference = Heterosexual)		
LGBTQ	1.75 **	2.81 ***
	[1.16,2.64]	[1.57,5.05]
*Former user vs. never*		
Sexual orientation (reference = Heterosexual)		
LGBTQ	0.95	2.82 ***
	[0.64,1.41]	[1.95,4.08]

** *p* < 0.01, *** *p* < 0.001.

**Table 3 healthcare-12-02083-t003:** Odds ratios in depression by smoking status among straight and LGBTQ participants.

	Full Sample	Heterosexual	LGBTQ
*Smoking status*			
Current user vs. never	1.24 [0.93,1.64]	1.25 [0.93,1.68]	1.08 [0.40,2.95]
Former user vs. never	1.00 [0.80,1.24]	1.02 [0.81,1.27]	0.85 [0.35,2.11]
*E-cigarette use status*			
Current user vs. never	2.07 ** [1.33,3.22]	1.94 ** [1.20,3.13]	3.67 * [1.06,12.74]
Former user vs. never	2.02 *** [1.54,2.66]	1.87 *** [1.39,2.50]	3.49 ** [1.52,8.00]

* *p* < 0.05, ** *p* < 0.01, *** *p* < 0.001. (Reference category: never smoked)

## Data Availability

The dataset supporting this article is available through the Health Information National Trends Survey (HINTS) 5, Cycle 4. (Hyperlink to dataset at https://hints.cancer.gov/default.aspx (Accessed on 12 March 2024)).

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
