# Peer review of "The Intersection of Sexual Orientation, Substance Use, and Mental Health: Findings from Hints 5"

_healthcare, 2024, doi:10.3390/healthcare12202083_

Round 1

Reviewer 1 Report

Comments and Suggestions for Authors

First of all, thanks for your work.

The paper it's well build, easy to read and globally well supported. The methods are well chosen and the results interesting.

I've some suggestions for you to improve your paper:

- The affirmation that you present in introduction between line 41 and 48 must be supported with references, because it's a strong affirmation.

- In the results you must refer again the test made when you show the numbers, it's easier for understand if you do it so.

- In discussion you must have some caution when you give some possible explanations for the results, like you do between line 179 and 181, because in this specific case economical aspects could also be an answer. I suggest that you review this kind of affirmations and support them with references if you needed.

I hope this little suggestions could improve the overall quality of paper.

Greetings

Author Response

First of all, thanks for your work.

The paper it's well build, easy to read and globally well supported. The methods are well chosen and the results interesting.

 Response: Thank you so much for taking the time to review our paper.

Comment 1: The affirmation that you present in introduction between line 41 and 48 must be supported with references, because it's a strong affirmation. 

 Response 1: We recognize that the affirmation presented in the introduction between lines 41 and 49 is a strong statement and requires proper support. To address this, we have revised the introduction to include relevant references that substantiate our claims. The updated references are now integrated into the text between lines 41 and 49 of the manuscript.

Comment 2: In the results you must refer again the test made when you show the numbers, it's easier for understand if you do it so.

Response 2: Agree. We have revised the Results section to include references to the specific statistical tests performed when presenting the numbers. This modification enhances clarity and helps readers understand the results by explicitly stating the methods used. The updated text can be found on lines 130-133, 143, 146, and 153 of the revised manuscript.

Comment 3: In discussion you must have some caution when you give some possible explanations for the results, like you do in this statement “This pattern might suggest that although e-cigarettes are initially attractive, their long-term use is not sustained, possibly due to emerging health concerns or the realization of their addictive nature” because in this specific case economical aspects could also be an answer. I suggest that you review this kind of affirmations and support them with references if you needed.

 Response 3:  Agree. We have accordingly revised the Discussion section to address concerns regarding the need for caution when providing possible explanations for the results. Specifically, we have acknowledged other contextual influences that may also contribute to the patterns observed. These changes can be found in the revised manuscript on lines 170-173, 180-182, 190-193, and 260-262.

Reviewer 2 Report

Comments and Suggestions for Authors

1.  Since sexual orientation is perhaps the key independent variable, it would be helpful to understand how it related to the other demographic variables being used as controls, even if this is placed as supplemental material.  This would help the readers better understand how controlling for the other demographics might be influencing the results.

2.  One possibility not mentioned in the discussion is that some factors involved in depression are no doubt transient losses (losing a job, a marriage breaking up, a pet dying, etc.).   People may turn to short-term solutions in such situations, like smoking or drug use.  However, with time, many people recover and no longer need/want the short-term solutions, so they become former users.  At the same time, some may become dependent on the drugs and become more depressed when they see how they have become dependent.  This is to say that there may be a reciprocal pattern between depression and drug use for some individuals. 

3.  One factor not investigated might be BMI and drug use.  I know of women who swore that smoking cigarettes helped them lose weight or keep it maintained at an acceptable level; when they quit, they gained weight.  Is there any relationship between BMI, cigarette use, and depression?

4.  There should be a mention of how other scholars may be able to access the data used here.

Author Response

Comment 1:  Since sexual orientation is perhaps the key independent variable, it would be helpful to understand how it related to the other demographic variables being used as controls, even if this is placed as supplemental material.  This would help the readers better understand how controlling for the other demographics might be influencing the results. 

Response 1: Agreed. We have provided a detailed supplemental analysis to illustrate better how sexual orientation relates to other demographic variables used as controls in our study. We have attached the supplemental material to help the readers.

Comment 2:  One possibility not mentioned in the discussion is that some factors involved in depression are no doubt transient losses (losing a job, a marriage breaking up, a pet dying, etc.).   People may turn to short-term solutions in such situations, like smoking or drug use.  However, with time, many people recover and no longer need/want the short-term solutions, so they become former users.  At the same time, some may become dependent on the drugs and become more depressed when they see how they have become dependent.  This is to say that there may be a reciprocal pattern between depression and drug use for some individuals.  

 Response 2: We appreciate your insight into the potential reciprocal relationship between depression and drug use, particularly in the context of transient losses and short-term coping mechanisms. We have revised the discussion to incorporate this perspective. The changes can be found between lines 256 and 262.

Comment 3: One factor not investigated might be BMI and drug use.  I know of women who swore that smoking cigarettes helped them lose weight or keep it maintained at an acceptable level; when they quit, they gained weight.  Is there any relationship between BMI, cigarette use, and depression? 

Response 3: Research indicates that smokers often have lower BMI and weight compared to non-smokers, partly due to nicotine's effects on metabolic rate and appetite. As a result, individuals may experience weight gain upon quitting smoking, though the extent of weight gain can vary between individuals.

There is also a notable association between smoking, obesity, and increased risk of anxiety and depression. For example, depression is significantly linked to smoking among obese women, and evidence suggests that higher BMI can be a contributing factor to depressive symptoms. The interplay between BMI, smoking, and depression is complex.

Comment 4:  There should be a mention of how other scholars may be able to access the data used here. 

 Response 4: We appreciate the suggestion. We have added a section in the manuscript detailing how other researchers can access the data used in this study. We have accordingly added in the access of data on between line 316 to 319.

Round 2

Reviewer 2 Report

Comments and Suggestions for Authors

1.  The authors' improvements are appreciated.  However, I detected a number of errors, especially in the improved sections.

2.  I was not able to read the supplementary files when I downloaded them, but I will trust the editors to make sure they are included when the paper is published.

3.  Line 64.  Suggest "outcomes with which LGBTQ share"

4.  Line 66.  Suggest "by sexual minorities which increase their"

5.  Line 70.  Suggest "use are associated with"

6.  Line 75.  I think you mean "for the LGBTQ community"?

7.  Line 89.  The average reader may not be aware if the HINTS 5 data were obtained from Canada or the USA or both, please clarify.

8.  Since the study was conducted at the start of the COVID pandemic when fear was rampant, some consideration should be given to how that situation may have enhanced or detracted from the study's generalizeability.

9.  Line 208, suggest "outcomes among LGBTQ indivduals"

10.  At lines 226 it is stated that education level was inversely related to smoking prevalence.  At line 197, the intersectionality of education/income sufficiency with sexual orientation is mentioned.  In light of these concerns, I would suggest testing for an interaction between education and/or income sufficiency and sexual orientation predicting smoking outcomes.  In general, education seems to reduce smoking outcomes, but is the effect of education moderated by sexual orientation or not? 

Comments on the Quality of English Language

As noted for the authors, there are a number of problems with the English in the revised sections.

Author Response

Comment 1: I was not able to read the supplementary files when I downloaded them, but I will trust the editors to make sure they are included when the paper is published.

Response 1: We will ensure that the supplementary files are correctly formatted and resubmitted for inclusion with the final paper.

Comment 2:  Line 64.  Suggest "outcomes with which LGBTQ share"

Response 2: We agree with the suggestion and have made the appropriate changes to line 64.

Comment 3:  Line 66.  Suggest "by sexual minorities which increase their"

Response 3: We have accepted this suggestion and revised line 66 to reflect the proposed wording: "by sexual minorities which increase their."

Comment 4:  Line 70.  Suggest "use are associated with"

Response 4: The suggested change has been incorporated. Line 70 now reads: "use are associated with."

Comment 5:   Line 75.  I think you mean "for the LGBTQ community"?

Response 5: Thank you for pointing this out. We have revised line 75 to now read "for the LGBTQ community."

Comment 6:  Line 89.  The average reader may not be aware if the HINTS 5 data were obtained from Canada or the USA or both, please clarify.

Response 6: We appreciate the comment. However, we would like to clarify that line 92 specifically mentions that the data were collected on the American public's knowledge.

Comment 7:  Since the study was conducted at the start of the COVID pandemic when fear was rampant, some consideration should be given to how that situation may have enhanced or detracted from the study's generalizability.

Response 7 : Thank you for this interesting point. It is possible that the COVID-19 pandemic had a limited impact on the study, which we highlighted in the limitation section as: Due to the unusual situation of the COVID-19 pandemic during the data collection period, our results may not apply to everyone, especially given the fact that pandemic-related may have stressors disproportionately affected different sexual orientation groups. Future research should take these factors into account and look at the long-term effects of sexual orientation and stress after the pandemic. 

Comment 8:   Line 208, suggest "outcomes among LGBTQ individuals"

Response 8: We appreciate the suggestion and have made the recommended change. Line 208 now reads "outcomes among LGBTQ individuals."

Comment 9:   At lines 226 it is stated that education level was inversely related to smoking prevalence.  At line 197, the intersectionality of education/income sufficiency with sexual orientation is mentioned.  In light of these concerns, I would suggest testing for an interaction between education and/or income sufficiency and sexual orientation predicting smoking outcomes.  In general, education seems to reduce smoking outcomes, but is the effect of education moderated by sexual orientation or not? 

Response 9: Thank you for this suggestion. We included the interaction terms between sex and education (sex*education) and sex and income (sex*income). However, the results did not show any significant interactions.
